# Discovering and Ranking Relevant Comment for Chinese Automatic Question-Answering System

**Siyuan Cheng [1], Didi Yin [1], Zhuoyan Hou [2], Zihao Shi [2], Dongyu Wang [2,\*] and Qiang Fu [1]**

[1] State Grid Hebei Electric Power Company Information & Telecommunication Branch, Shijiazhuang 266590, China

[2] College of Artificial Intelligence, Beijing University of Posts & Telecommunications, Beijing 100089, China

\* Correspondence: dy_wang@bupt.edu.cn; Tel.:+86-133-1105-0257

**Abstract:** Intelligent customer service system is timely, efficient, and accurate, which is more and more popular in grid electric power companies, and the amount of customer consultation is increasing day by day. It is infeasible for human customer service to answer these questions on time, so an automatic question-answering system is of great help to the grid electric power company. The customer queries from the grid electric power company customer service is very different from open-domain questions: the problems questioned by customer tend to be for a specific device or system within the enterprise operation problem. Most grid electric companies provide customers with a communication platform where customers can get guidance on using equipment and the business process. The comments from communication platforms are valuable resources for answering customer questions. In our work, we use three neural network models which excavate potential answers to customer queries from comments. One of the key challenges, however, is the difficulty of matching customer questions with comments. To solve this problem, we propose a method based on deep learning to find the comments related to customer questions to generate more accurate and reliable answers. Experiments can prove that our method performed well in the customer service of grid electric power company.

**Keywords:** deep learning; customer service system; automatic question-answering

## 1. Introduction

With the advent of the information society, artificial intelligence technology and natural language processing technology are rapidly developing, and intelligent services are used in depth in the field of grid electric power companies. Because of the large number of questions and the complex range of queries, customer service cannot answer all customer questions on time. Most grid electric companies provide customers with a communication platform where customers can get guidance on using equipment and the business process. To answer customer questions timely and accurately and promote customer satisfaction, we propose a solution to find the answers to customer queries from the comments already published on the communication platform.

To illustrate how useful the comments might be, two examples are shown in Table 1. There are a customer query, a ground-truth answer, and a relevant comment written by the customer. Consider the question, "The printer in the office cannot work and no paper is coming out. What should I do?" The ground-truth answer to the question is "Remove the paper from the printer and then reuse it." We can find this answer from the comment "There is a paper jam in the public printer, which locates on the ninth floor of the main building. After taking out the paper, it can be used normally after debugging.", which has been posted on the platform earlier. Therefore, the comment is relevant to the answer to the customer's query. It helps discover the real answer to find the relevant comment.

**Table 1.** Answer and comment for two customer queries.

| |
|---|
| **Question**: The printer in the office cannot work and no paper is coming out. What should I do? **Answer**: Remove the paper from the printer and then reuse it. **Comment**: There is a paper jam in the public printer, which locates on the ninth floor of the main building. After taking out the paper, it can be used normally after debugging. |
| **Question**: I forgot the computer password. I have inputted the password too many times. Now the computer won't allow me to input it. How can I solve this problem? **Answer**: Shut down for 20 30 minutes, and then log on again. **Comment**: There are too many times to enter the power-on password, so the computer is locked. Turn it off for about 20 minutes and then turn it on again. |

Traditional question answering system always uses word-based relevance function. Additionally, before inputting into the neural model, the text data usually rely on manual feature engineering [1]. In recent years, with the continuous development of deep learning technology, the DL-based (Deep Learning-based) question-answering system has gradually become mainstream [2]. Ref. [3] proposed a framework that can return an answer to a customer question by extracting a sentence from reviews or a question-answering pair. Ref. [4] applied a multi-task deep learning method with attention mechanisms to analyze the relevance between the answer and the reviews that might become answers, exploiting many customer-generated question-answering data. Ref. [5] proposed a framework named Riker with interpretability and effectiveness. The state-of-art Riker model can mine rich keyword representations of customer queries. The Google team proposed the Transformer model [6], which extensively used self-attention mechanisms. Moreover, the transformer-based model like BERT (Bidirectional Encoder Representation from Transformers) [7], T5 (Text-to-Text Transfer Transformer) [8], Deberta (Decoding-enhanced-BERT-with-disentangled attention) [9] are applied to several NLP tasks. This transformer-based model has also been widely applied in question-answering systems. Ref. [10] proposed two neural models, which find relevant reviews for the customer query.

At present, the research on a question-answering system to answer customer questions can be roughly divided into two categories: one is extractive method [1,3,11,12], and the other is abstractive method [13,14]. On the one hand, the extractive method mainly extracts useful fragments from relevant reviews to organize the answer to a question. Ref. [11] incorporated aspects of personalization and ambiguity, which has extended the extractive method. Refs [12] proposed the learned model that learns latent aspects and aspect-specific embeddings of reviews to predict answers. On the other hand, the abstract methods mainly generate the predicted answer to the question from the vocabulary list. Ref. [13] proposed an adversarial learning-based model, which is composed of a review representation module, a key-value memory network, and a recurrent neural network. Ref. [14] proposed a model RAGE to generate more accurate and informative answers with human language.

The above research achievements are mostly applied in the field of e-commerce customer service, for example, product-related question answering and community question answering. Most of them are intended for specific simple scenarios and have domain uniqueness, which cannot be directly applied to the non-public grid electric power company field.

However, since there is no direct connection between customer queries and comments from communication platforms, it is difficult to train a supervised system to find relevant comments for customer queries. The busy customer service system leads to a large number of comments for guidance on using equipment as well as the business process. Therefore, creating an annotated dataset for the relevant connection between the customer queries and the user comments is very complex and difficult, which is laborious work and results in inefficiency.

To improve efficiency and reduce cost, based on the content of the grid electric power company customer service dialogue and the comments of the customers on the communication platform, we proposed a better way to solve this problem. We used the existing

question-and-answer pairs as the supervised signals, took the answer prediction as the optimization objective, and decomposed the problem into two sub-tasks: one was to match the question and the comments from the communication platform, and the other was to match the comments from the communication platform and the answer.

To sum up, the contributions of this work can be summarized as follows:

- We verify the performance of three neural network models in question-answering tasks under the scenario of grid electric power company customer service. By predicting the answer to customer queries through the content of comments from platforms, the ultimate goal of the model is to find useful comments that can provide useful answers to customer queries.
- The input of our end-to-end neural network models is raw texts, which do not need to be processed by feature engineering before inputting into the model.
- We further build a dataset based on the data provided by the State Grid Hebei Electric Power Company and use them to construct a question-answering system suitable for the grid electric power field.

In this paper, we will first give a brief introduction of the background as well as the main contributions of our work. Following this, we will introduce the academic work on automatic question-answering systems. Then we are going to analyze and dig out the relevance between the question and comments, and there are three models shown (i.e., NN-RC, Bert-RC, Deberta-RC) in the third section. The fourth section discusses the experimental results and compares the performance of different models. Finally, we will summarize all the work in our paper and look forward to future work.

## 2. Related Work

According to iiMedia [15] Research, the scale of China's intelligent customer service industry reached 150 billion yuan in 2020 and is expected to reach one trillion yuan in 2030. The intelligent customer service system combines various technologies such as natural language understanding, knowledge management, and automatic question-answering system. As an enterprise-oriented project, the intelligent customer service system connects enterprises with users by natural language processing and provides enterprises with statistical analysis information needed for fine management [16]. In recent years, more and more companies have been paying attention to intelligent customer service [17]. At present, in the public domains of community and e-commerce, such as Alibaba Timi [18], Tecent Qidian [19], Sobot [20], and so on, are widely used in China. Compared with manual customer service, the application of intelligent customer service greatly improves the response rate, and the speed of information construction [21].

According to the data and information provided by the state grid Hebei electric power company information and the telecommunication branch, employees of the grid electric power company always get help from customer service when they encounter problems that need to be solved. Due to the concentrated working time, the customer service hotline will face peak hours during working hours. The time the customer service spends to solve problems is uncertain, ranging from a few minutes to a few hours or even a day, so the employee cannot obtain the solution immediately, which affects the work progress and reduces the efficiency of the employee.

At present, there are few types of research on intelligent customer service for grid electric power companies. Ref. [22] obtained the emotion classification model based on the bidirectional long-term memory network training. The problem-solving needs are divided into three levels, and the classification results can be used in the automatic priority scheduling decision to judge whether to access the human service immediately. Ref. [23] refined the design of the online power intelligent customer service system and explored the design strategy of the online power intelligent customer service system based on artificial intelligence, aiming to provide new ideas for intelligent customer service system designers. Ref. [24] studied the current application status of artificial intelligence and analyzed the future development trend, and put forward the specific application scenarios of artificial in-

telligence in the fields of intelligent dispatching, intelligent inspection, equipment essential security, customer service, and so on. Ref. [25] proposed a knowledge base construction method based on a knowledge graph and designed the technical scheme of an intelligent customer service system by combining a customer service knowledge base and knowledge retrieval technology based on the graph. Ref. [2] proposed the SoftLexicon + Bi-LSTM-CRF model in NER tasks under a KBQA scenario, which fills the gap in the research of intelligent customer service-related technologies in the power grid field in China.

## 3. Method and Approach

When we receive a customer query, we want to find relevant comments that might provide the correct answer to the query. Because we do not have annotated query and comments pairs, that is, we do not have data that label comments as relevant or irrelevant comments for customer queries, we cannot directly train classification or ranking models to label or rank comments for specific customer queries.

However, by mining existing customer questions that have already been answered by human customer service of the gird electric power company, we can get many labeled question-answering pairs. To find the relevance between the customer query and comments from communication platforms, we use the existing question-answer supervision signals to build a model. The goal of the model is to calculate the probability that a potential comment is relevant to the answer to a customer question. The formula is as follows:

$$P(a \mid q) = \sum_{c_i} P(c_i \mid q) P(a \mid c_i, q) \tag{1}$$

where $a$ represents the answer, $c$ represents a list of comments sentences, $c_i$ represents each comment sentence, and $q$ represents the customer question.

In this formula, the relevance function between the answer and customer question ($P(a \mid q)$) is decomposed into two relevance functions: one is the relevance function between customer query and comment ($P(c_i \mid q)$), another is the relevance function between comment and answer ($P(a \mid c_i, q)$). It should be noted that the implicit premise of this formula is that we assume that the list of comments we use is useful in predicting the most relevant answer for each specific customer question; that is, no customer question cannot be answered.

In our work, we treat it as a binary classification problem. We use the different kinds of neural networks parameterized by the two relevance functions $P(c_i \mid q)$ and $P(a \mid c_i, q)$. In the experiment, the goal of our neural network model is to rank answers to customer questions. The optimization of our neural network model is scoring real answers higher than the wrong answers randomly selected from other answers.

In our experiments, we are not to calculate a set of answers on the probability distribution ($P(a \mid q)$) but calculate a bounded score (0 or 1) for an answer ($S(a \mid q)$). Based on a margin loss, the loss function as shown below:

$$loss = \max(0, S(a \mid q) - S(a' \mid q) - \delta) \tag{2}$$

where a represents the correct answer, $a'$ represents the wrong answer randomly selected from other answers, and $\delta$ is the margin hyperparameter.

As can be seen from the formula, the training process of our model is to score a real answer. The difference between the true answer and the wrong answer chosen at random is at least $\delta$.

### 3.1. NN-RC

In the experiment, Fasttext [26] is used to encode the sentences. As shown in Figure 1, there are three layers in the Fasttext model architecture: the input layer, the hidden layer, and the output layer. Because of the simple structure, the training time of the model is greatly reduced. In the previous work, the Fasttext model is generally used for the text

classification task, but the model also generates word vectors when carrying out the text classification task; that is, word vectors are the by-products of the text classification task. The model can generate a vector representing the document from the output layer to the hidden layer.

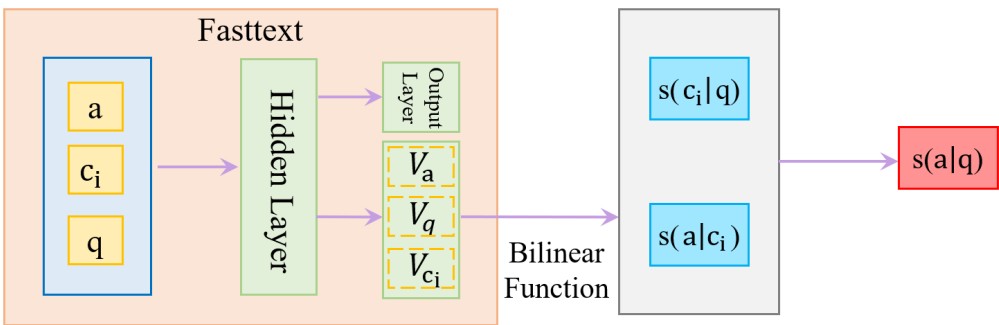

**Figure 1.** Model architecture of NN-RC.

In NN-RC, we obtain the vector representations $v_a$, $v_q$, and $v_{c_i}$ for $q$, $a$, and $c_i$ by calculating the average value of pre-trained word embedding representations in sentences. After obtaining the embedding representation of the sentence, we use a bilinear function to model the two relevance functions, as shown in the following formula:

$$S(c_i \mid q) = \sigma\left(v_q W_1 v_{c_i}^T\right) \tag{3}$$

$$S(a \mid c_i) = \sigma\left(v_{c_i} W_2 v_a^T\right) \tag{4}$$

where $W_n (n \in 1, 2)$ are model parameters.

Based on the above two formulas, we scored the answers, the formulas are shown following:

$$S(a \mid q) = \sum_{c_i} S(a \mid c_i) S(c_i \mid q) \tag{5}$$

where $c$ represents all the comments that can answer the question $q$.

In this model, we set the sum of cumulative confidence as 1.0 (i.e., $\sum_{c_i} S(c_i \mid q) = 1.0$), and in the experiment, we use the sigmoid function to make the constraint of this confidence scores.

*3.2. Bert-RC*

In Bert-RC, we use the transformer-based model Bert [7], a pre-trained model over a massive corpus. In essence, the vector representation of the words is calculated from the context and can represent the polysemy of the words, so the semantic representation ability of sentences is enhanced. Bert adopts a bidirectional transformer as a feature extractor and adopts a self-attention mechanism to model the text. In Bert, the correlation between each word and another word in the sentence is calculated [27]. The correlation between words reflects the relevance and importance of different words. Then, these relationships are used to adjust the importance (i.e., weight) of each word, and a new semantic representation is learned, which includes not only the word itself but also the relation between the words. Compared with simple word vectors, BERT can learn more comprehensive semantic expressions.

In this model, as shown in Figure 2, we use Bert directly caculate the relevance function ($S(c_i \mid q)$ and $S(a \mid c_i)$), the formulas are shown following:

$$S(c_i \mid q) = \text{softmax}\left(W_1^T \text{ BERT}(c_i, q)\right) \tag{6}$$

$$S(a \mid c_i) = \sigma\left(W_2^T \text{ BERT}(a, c_i)\right) \tag{7}$$

Based on bert, Bert-RC can perform a more fine-grained analysis on two pairs of texts; for example, the attention mechanism of bert can evaluate the similarity of all markers between text pairs.

As with NN-RC, we set the sum of the cumulative confidence as 1.0 (i.e., $\sum_{c_i} S(c_i \mid q) = 1.0$). However, in this experiment, we use the softmax function to make the constraint of these confidence scores.

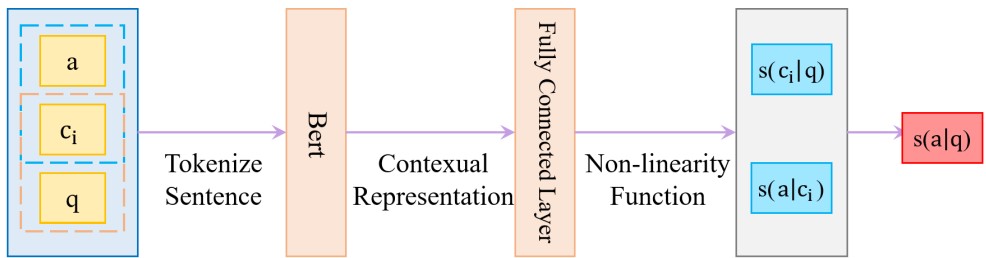

**Figure 2.** Model architecture of Bert-RC

### 3.3. DeBerta-RC

In DeBerta-RC, we use DeBerta [9], a state-of-art pre-trained language model that performs well on the NLP tasks. DeBerta enhances the dependence between location and content by increasing the self-attention of location content and content-location and uses EMD (enhanced MASK decoder) to alleviate the mismatch problem caused by mask in BERT pre-training and fine-tuning.

As shown in Figure 3, DeBerta-RC shares the same architecture as Bert-RC. The difference between Bert-RC and DeBerta-RC is that DeBerta-RC computes the relevance functions ($S(c_i \mid q)$ and $S(a \mid c_i)$) directly with Deberta, the formulas are shown following:

$$S(c_i \mid q) = \text{softmax}\left( W_1^T \, \text{DeBerta}(c_i, q) \right) \tag{8}$$

$$S(a \mid c_i) = \sigma\left( W_2^T \, \text{DeBerta}(a, c_i) \right) \tag{9}$$

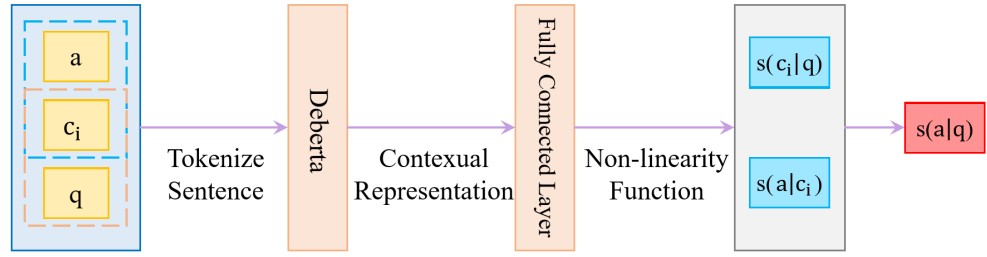

**Figure 3.** Model architecture of DeBerta-RC

### 3.4. Comment Filtering

In our experiment, each comment is broken down into individual sentences, representing a sentence in each comment rather than the complete comment. However, a large amount of comments is considered for each business type, resulting in a large computational cost.

To solve the above problems, we preprocess the comment. We pre-rank the comment that might be relevant to the customer's query and then clear the comments that obtain the lowest score before inputting them into the model.

In this work, we use a simple bert classifier. We first fine-tune the pre-trained Bert on question-answer pairs to predict whether an answer is a true answer to the user's query. When training the neural network model, the previously fine-tuned Bert is selectively applied to question-comment pairs to obtain the pre-ranking of the comments. We select

the top n comments from the pre-ranking model and only select the top n comments to input into the model for the training process [28]. In this paper, when applying the filtering, we denote model A as A + FLTR; for example, NN-RC + FLTR means that NN-RC applies the comment filtering.

### 3.5. Cross-Domain Pre-Training

Based on the data provided by the grid electric power company, we created datasets that contain different categories. In our experiment, every category serves as an independent domain. When training the model, we train a number of separate models in every domain. However, there are many similar customer queries in different categories. We pre-train a general model that combines data from all categories and then fine-tuning it for each category. In this paper, when applying the cross-domain pre-training, we denote the model A as A + PRE; for example, NN-RC + PRE means the NN-RC is first pre-trained with the data from all categories and then fine-tuned for each category.

## 4. Results and Discussion

### 4.1. Data Introduction

We used the data provided by the grid electric power company and developed the dataset for the experiment. As shown in Table 2, the data are divided into three categories according to the type of service: communication, business application, terminal and peripheral. The communication includes the installation and movement of telephone, transferring calls, telephone number queries, and so on; the business application includes the first-level deployed information system, the second-level deployed information system, the self-built information system, and so on; The terminal and peripheral includes a fax machine, printer, duplicator, scanner and so on. We divide the data into the training set, development set, and test set and divide the specific content of comments into sentences. Therefore, each comment is actually a comment sentence rather than a complete comment from a customer. From the perspective of the application of the customer service system, it is more convenient and friendly to present the sentence related to the answer after receiving the customer's queries than to present the complex and complete comments from other customers.

**Table 2.** Data set statistics.

| Category | Questions [1] | Services [1] | Comments [1] |
|---|---|---|---|
| Communication | 9256 | 342 | 200,278 |
| Business Application | 6386 | 157 | 130,964 |
| Terminal & Peripheral | 8832 | 286 | 154,394 |

[1] represents the number.

### 4.2. Qualitative Evaluation

#### 4.2.1. Evaluation Metrics

Since there is no annotated query-comment data, to evaluate the model quantitatively, we introduce AUC [29] evaluation index; the formula is as follows:

$$\frac{1}{|Q|} \sum_{q \in Q} \frac{1}{|A|} \sum_{a' \in A} S(a > a') \tag{10}$$

where a is the real answer, $a'$ is the wrong answer chosen randomly from other answers, $A$ is the set of $a'$, and $Q$ is the set of all customer queries.

AUC can calculate the proportion of the real answer to each customer query that is assigned a higher score by the model we are using. When the calculated result is 1.0, the accuracy of the model is 100%.

At the same time, we use the model MOE [30] as the baseline system for comparison. The MOE model is trained by a large corpus of questions that have been answered to learn the relevance function. The traditional model MOE studies the relevance function based on

vocabulary level and applies the artificial standard feature engineering to process the data before inputting them into the model.

### 4.2.2. Experiment Details

When applying the MOE, we use open source in the experiment, and the parameters are the default optimal configuration for the model. When applying the comment filter, we only kept the top 10 comments in the experiment. When applying the two different neural network models (i.e., NN-RC and Bert-RC), we set the upper limit of the total number of comments to 100 in the experiment, and then we delete the rest answers ranking beyond 100.

### 4.2.3. Result and Disscusion

As can be seen from Table 3, the baseline (MoE) performs better than NN-RC in all categories. When applying cross-domain pre-training, NN-RC + PRE outperforms the baseline model. The cross-domain pre-training improves the model, obviously. However, when we applied the filtering and kept only the top-10 comments, there was a drop in all categories because some comments ranked low may still provide information relevant to the customer's problem. We continued to increase the number of commets to keep (from the top 10 to the top 50) but found that the model with the filter still performed worse than NN-RC + PRE.

Due to memory constraints, we did not run Bert-RC and DeBerta-RC with all comments. As can be seen from Table 4, the performance of Bert-RC is significantly outstanding than baseline as well as all the NN-RC models (NN-RC & NN-RC + PRE & NN-RC + FLTR+ PRE). Even when the filter is applied, Bert-RC + FLTR still performs better than NN-RC, with an average performance improvement of about 11% over NN-RC and about 7% over NN-RC + FLTR + PRE. When we pre-train it in a different domain, the Bert-RC + FLTR + PRE model significantly outperforms the NN-RC model and baseline. The strong performance of the Bert-RC model is likely attributed to the self-attention mechanism, which can conduct a more fine-grained analysis of words in sentences.

Look at the DeBerta-RC. Encouragingly, even with the comments filtering, DeBerta-RC has a superior performance over the other models. There is no doubt that this model is the best-performing model in our experiments. Applying pre-training improves the model further, with the largest gain in Communication. We hypothesize that the disentangled attention contributes to the outstanding performance of DeBerta-RC.

**Table 3.** AUC performance of models.

|  | **Baseline** | **NN-RC** | **NN-RC + PRE** | **NN-RC + FLTR + PRE** |
|---|---|---|---|---|
| Communication | 0.7073 | 0.6864 | 0.7518 | 0.7249 |
| Business Application | 0.7133 | 0.6890 | 0.7477 | 0.7191 |
| Terminal & Peripheral | 0.6925 | 0.6836 | 0.7538 | 0.7321 |
| **Average** | 0.7044 | 0.6863 | 0.7511 | 0.7254 |

**Table 4.** AUC performance of models.

|  | **Bert-RC + FLTR** | **Bert-RC + FLTR + Pre** | **DeBerta-RC + FLTR** | **DeBerta-RC + FLTR + Pre** |
|---|---|---|---|---|
| Communication | 0.8011 | 0.8257 | 0.8716 | 0.9135 |
| Business Application | 0.7977 | 0.8138 | 0.8578 | 0.8918 |
| Terminal & Peripheral | 0.8035 | 0.8272 | 0.8707 | 0.9052 |
| **Average** | 0.8008 | 0.8222 | 0.8687 | 0.9035 |

### 4.3. Qualitative Survey

In the last subsection, we evaluate how well different models find the relevant comments. However, we have not tested the actual utility of their comments. To this end, we

invite annotators to score how relevant the comments are to the answer. Because the model named DeBerta-RC + FLTR + Pre outperforms the other models in the last subsection, we take this model as an example.

We randomly selected 200 questions from the three categories (i.e., Communication, Business Application, and Terminal & Peripheral). All the annotators rate how well/helpful a commment answers the question from the customer and give a score $\alpha$ ($\alpha \in [0,3]$), where 0 indicates that comment is not relevant to the question and 3 indicates that comment directly answers the question. The final score for each comment is the average score given by five annotators.

The distribution of the helpfulness scores is displayed in Figure 4. Interestingly, the results are very complicated; we can see that not all of the comments are helpful, suggesting that there is still plenty of room for improvement.

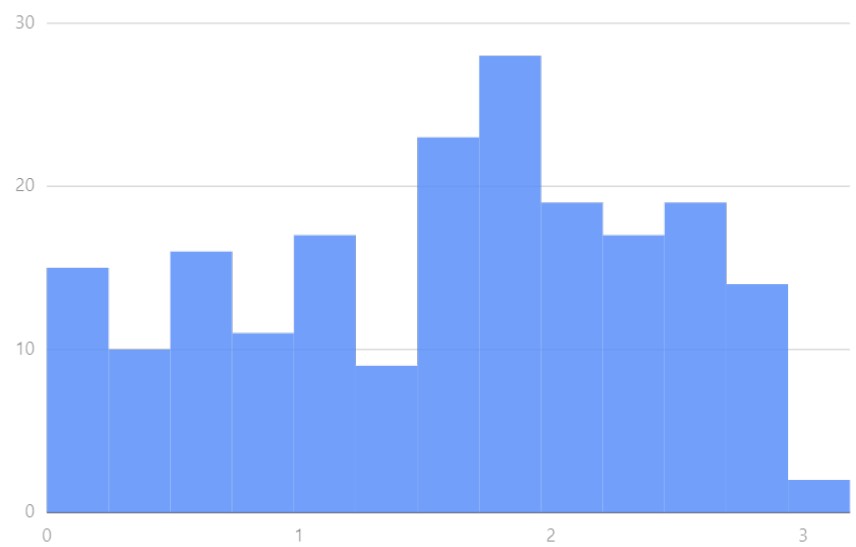

**Figure 4.** Distribution of average helpfulness score. (Mean = 1.58, std = 0.82).

### 5. Conclusions

Discovering relevant comments for customer queries is significant to establishing grid electric power companies. Because of the lack of annotated comments, it is a challenging problem to find useful comments for customer queries. In this paper, we adopt a method that leverages comments to predict answers to customer queries. We present three neural models, one is based on the simple neural network model (NN-RC), another is a model based on Bert (Bert-RC), and the other is an advanced model based on DeBerta (DeBerta-RC). The criterion for evaluating the model is the ability to predict answers and find relevant comments. The performance of the three neural network models is compared with the performance of the baseline model. The final results show that all the models can get satisfactory results, and the model based on DeBerta produces the best performance.

There is currently little effective research on grid electric power companies' intelligent customer service systems. The automatic question-answering system we produced will fill the gap in the research of intelligent customer service system technology in the field of grid electric power. The method we proposed still has some shortcomings and limitations. For example, in practice, not every customer query can be answered by comments: the existing set of comments may not contain any relevant answers for customer queries.

As we have seen from Section 4.3, many questions may not be answered with limited comments. When a question cannot be answered, the system should return zero answers instead of providing a list of irrelevant comments, which can have a negative impact on the customer experience. In the future, we will focus on the problem of answer reliability

for the automatic question-answering system in grid electric power companies in Chinese. Based on the work in this paper, we are going to propose a rejection model to reject unreliable answers.

**Author Contributions:** Conceptualization, S.C. and D.Y.; methodology, S.C., D.Y. and Z.S.; software, D.Y. and Z.H.; validation, S.C., D.Y. and Z.H.; formal analysis, D.Y. and Z.H.; investigation, Z.S., D.W. and Q.F.; resources, S.C., D.Y. and Q.F.; data curation, S.C., D.Y. and Q.F.; writing original draft preparation, Z.H.; writing review and editing, S.C.; visualization, D.Y.; supervision, Q.F.; project administration, Z.S., D.W. and Q.F.; funding acquisition, Q.F. All authors have read and agreed to the published version of the manuscript.

**Funding:** This research was funded by the science and technology project of State Grid Hebei Electric Power Company Information & Telecommunication Branch "Research on semantic retrieval and analysis technology of intelligent customer service based on deep learning" (SGHEXT00DDJS2100105).

**Institutional Review Board Statement:** Not applicable.

**Informed Consent Statement:** Not applicable.

**Data Availability Statement:** Data are available upon request.

**Acknowledgments:** We thank State Grid Hebei Electric Power Company Information & Telecommunication Branch for providing the research data and funding support for this paper.

**Conflicts of Interest:** The authors declare no conflict of interest.

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
