# Peer review of "Discovering and Ranking Relevant Comment for Chinese Automatic Question-Answering System"

_applsci, doi:10.3390/app13042716_

Round 1

Reviewer 1 Report

The paper proposes a deep neural network-based Q&A system to generate answers in a sentence to a given question, which takes pairs of questions and relevant answers found by external mining techniques. The main idea is, I guess, to train language models such as Fasttext (for baseline), BERT and DeBerta with the loss function derived from a topic-model like likelihood function shown in Equation (1).

If I understood correctly, the idea of the paper is novel and very interesting. I am not so sure, however, that I got the point correctly because the main idea is not presented kindly and clearly in the manuscript. Furthermore, more extensive experiments should be given to evaluate the performance of their great idea. The quality of figures such as Figure 1, 2, and 3 also should be improved.

While the idea is novel and promising, the manuscript could benefit from better presentation and more extensive experiments. The figures could also use some improvement in quality. With improved representation, this idea could be suitable for publication.

Reviewer 2 Report

The authors presented an interesting idea for discovering and evaluating comments for the Chinese automatic question answering system. As the number of customer consultations is increasing day by day, the intelligent system of answering questions is very helpful both for the power grid and in other service areas. The method uses comments from communication platforms that provide imprecise answers to customer questions. To match customer questions with comments, a deep learning model was proposed to find comments related to customer questions and then designate more precise and reliable answers.

This legitimate method, however, requires the description in the manuscript to be clarified before it can be published.

1. In the Introduction, add a paragraph about the organization of the article (What is in each section?). However, the last Conclusions section lacks a short description of Future Work.

2. In the equation (1), variables are supposed to be written by italics. In the equation (10), 1/Q => 1/|Q|

3. The dataset is described very casually and briefly. There is no citation of the web site from which you can download the collection and repeat the research results. In addition, Table numbers start with number 2.

Minor typos:

line 40: mainstream[24] => mainstream[24]

lines 47-50: The following sentence is unclear: Moreover, the transformer-based model like BERT (Bidirectional Encoder Representation from

Transformers) [6], T5 (Text-to-Text Transfer Transformer) [7], Deberta (Decoding-enhanced-

BERT-with-disentangled attention) [8].

lines 104-105: The following sentence is also unnecessary: More and more artificial intelligence algorithms are applied to real life.

line 300: Conclusions => Conclusions.

Once these inaccuracies have been corrected, the manuscript can be published.
